# Electron diffraction of deeply supercooled water in no man's land

Constantin R. Krüger [1,2], Nathan J. Mowry[1,2], Gabriele Bongiovanni[1,2], Marcel Drabbels [1] & Ulrich J. Lorenz [1] ✉

A generally accepted understanding of the anomalous properties of water will only emerge if it becomes possible to systematically characterize water in the deeply supercooled regime, from where the anomalies appear to emanate. This has largely remained elusive because water crystallizes rapidly between 160 K and 232 K. Here, we present an experimental approach to rapidly prepare deeply supercooled water at a well-defined temperature and probe it with electron diffraction before crystallization occurs. We show that as water is cooled from room temperature to cryogenic temperature, its structure evolves smoothly, approaching that of amorphous ice just below 200 K. Our experiments narrow down the range of possible explanations for the origin of the water anomalies and open up new avenues for studying supercooled water.

Water has been called "the most anomalous liquid"[1], with over 70 anomalous properties that have been identified to date[2]. Several competing theories have been put forward to explain the origin of these anomalies[3,4]. The liquid-liquid critical point scenario posits that at low temperatures, supercooled water exists in a high and a low density phase, with the phase coexistence line terminating in a critical point[5,6]. In this model, the anomalies manifest as water approaches the Widom line that emanates from this critical point[7]. The critical-point-free scenario similarly proposes the existence of a liquid-liquid phase transition in the supercooled regime, but with the critical point at negative pressure[8,9]. Other theories explain the anomalies without requiring the existence of a singularity[10]. Most vexingly, the experimental verification of these theories has largely remained elusive because of the fast crystallization of water in the temperature range of 160–232 K, frequently nicknamed "no man's land"[3]. X-ray diffraction of evaporatively cooled microdroplets has revealed a smooth evolution of the structure factor down to 227 K (ref. 11), but has been unable to access lower temperatures due to rapid crystallization. Infrared spectra of transiently heated amorphous ices are consistent with a two state-mixture of a high and a low temperature motif in no man's land[12]. However, the approach does not probe the liquid itself, but rather an amorphous ice that has sampled a range of temperatures. A definitive explanation of the origin of the water anomalies can only emerge if water can be systematically characterized throughout no man's land. This requires preparing the supercooled liquid at a well-defined temperature and probing it directly before crystallization occurs. Here, we present an experimental approach that overcomes these challenges and allows us to capture the structural evolution of water as it is cooled from room temperature to cryogenic temperature.

## Results

Experiments are performed with a time-resolved electron microscope developed in our laboratory (Supplementary Methods A-D)[13,14]. The sample geometry and experimental concept are illustrated in Fig. 1. A 600 mesh gold grid (Fig. 1a) supports a holey gold film (2 µm holes) that is covered with a sheet of few-layer graphene (Fig. 1b). The sample is cooled to 101 K, and a 176 nm layer of amorphous solid water is deposited in situ. In order to prepare water in no man's land, we locally heat the sample with a shaped microsecond laser pulse, with the laser beam centered onto one of the squares of the specimen grid (532 nm wavelength). We then use an intense, high-brightness electron pulse (200 kV accelerating voltage) to capture a diffraction pattern of the supercooled liquid (Fig. 1c).

Figure 2a illustrates the typical shape of the microsecond laser pulse (green) that we use to prepare water in no man's land, with the simulated temperature evolution of the sample shown in black (Supplementary Methods E). We first heat the sample to room temperature with a 30 µs laser pulse, before reducing the laser power in order to rapidly cool the liquid to a well-defined temperature in no man's land. This sequence maximizes the observation time that is available before

[1]Ecole Polytechnique Fédérale de Lausanne (EPFL), Laboratory of Molecular Nanodynamics, CH-1015 Lausanne, Switzerland. [2]These authors contributed equally: Constantin R. Krüger, Nathan J. Mowry, Gabriele Bongiovanni. ✉e-mail: ulrich.lorenz@epfl.ch

### Sample geometry and experimental approach

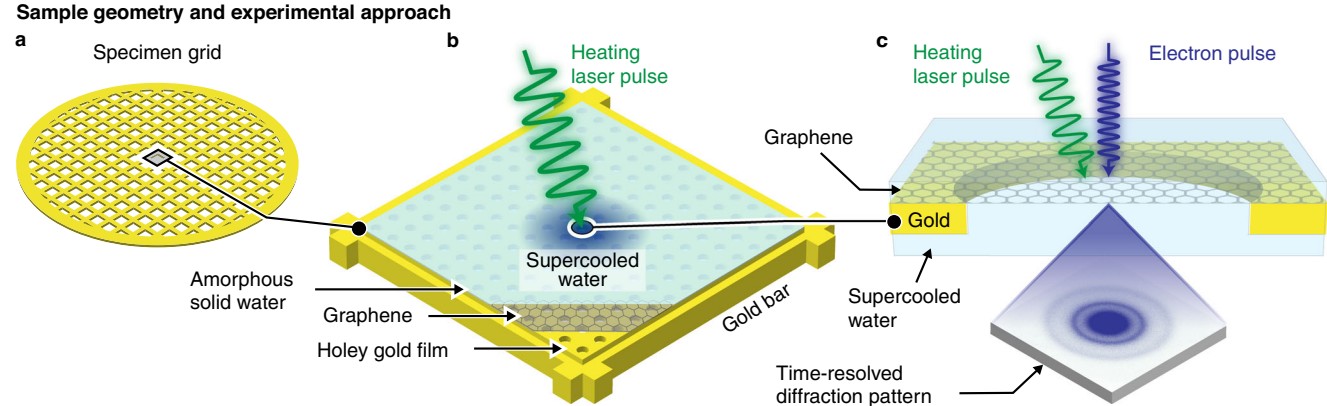

**Fig. 1 | Illustration of the experimental approach. a**, **b** Illustration of the sample geometry. A gold mesh supports a holey gold film that is covered with few-layer graphene. A 176 nm thick layer of amorphous solid water is deposited (101 K sample temperature), which is then locally heated with a shaped microsecond laser pulse to prepare water in no man's land. **c** A diffraction pattern of the supercooled liquid is captured with an intense, 6 μs electron pulse.

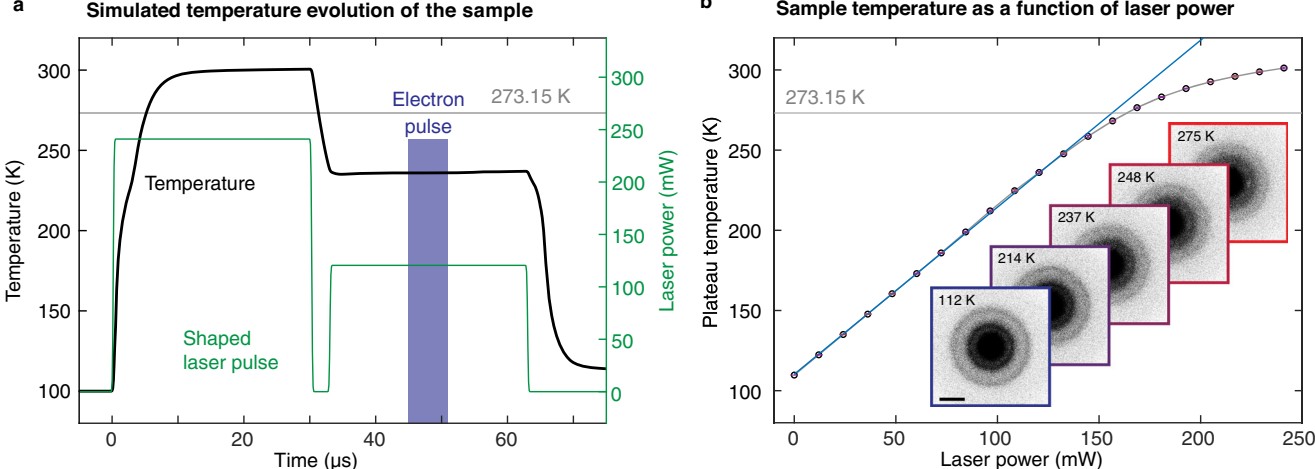

**Fig. 2 | Simulation of temperature evolution of the sample. a** Simulation of the temperature evolution of the sample (black) under irradiation with a shaped microsecond laser pulse (green). The sample is first heated above the melting point and then rapidly cooled to the desired temperature in no man's land by reducing the laser power. Once the temperature has stabilized, we capture a diffraction pattern with a 6 μs electron pulse (blue). **b** Simulations show that this temperature (black circles) increases linearly with laser power. It only starts to level off above ~260 K, where evaporative cooling becomes important. The black line corresponds to a spline of the simulated data points, while the blue line is a linear fit. The inset shows diffraction patterns recorded over a range of temperatures. Scale bar, 2 Å⁻¹.

crystallization sets in. In contrast, if the sample is simply heated up to reach no man's land, it crystallizes rapidly, since it first has to pass through the temperature regime near 200 K, where the nucleation rate has a maximum[3,15]. In the example simulation shown in Fig. 2a, the laser power is reduced by half, which causes the liquid to rapidly cool from 300 K to 236 K. At a delay of 15 μs after reducing the laser power, we then capture a diffraction pattern of the supercooled liquid with a 6 μs electron pulse before crystallization sets in. Note that in order to reduce the cooling time, we initially lower the laser power even further, as shown in Fig. 2a.

As detailed in Supplementary Notes H, we confirm that the temperature of the sample has stabilized when we probe its structure by characterizing the cooling process with time-resolved electron diffraction. Even in the temperature range where the cooling is expected to be slowest, we measure a 1/e cooling time of only 11 μs. The fast cooling is a consequence of the close proximity of the area under observation to the bars of the specimen grid, which remain at cryogenic temperature throughout the experiment and therefore act as an efficient heat sink[16–18].

The specific heat transfer properties of the sample geometry make it straightforward to determine the temperature at which the liquid stabilizes for a given laser power (Supplementary Methods F).

Simulations show that this temperature increases linearly with laser power and only starts to level off above ~260 K, where evaporative cooling becomes important (Fig. 2b). This allows us to determine the sample temperature as follows. By comparing with x-ray data in the mildly supercooled regime[11], we obtain a temperature calibration of our experiment at intermediate laser powers. At zero laser power, the sample temperature only slightly exceeds the temperature before the laser pulse. We can therefore use simulations to determine the temperature at zero laser power with good accuracy. For all other laser powers in the linear regime (which includes no man's land), we interpolate linearly. Data recorded at higher laser powers are corrected for the effect of evaporative cooling as described in Supplementary Methods F.

Figure 3a shows the temperature evolution of the diffraction pattern of water, revealing that the structure of water evolves smoothly as the liquid is cooled from 290 K to 180 K (Supplementary Methods G). This is also evident in the two-dimensional plot of Fig. 3b, where the positions of the first two diffraction maxima are indicated with black dots. The gray lines represent splines that provide a guide to the eye. As shown in Supplementary Fig. 10, the positions of the diffraction maxima are largely consistent with x-ray data that are available for temperatures above 227 K (Ref. 11, 19). The position of the second

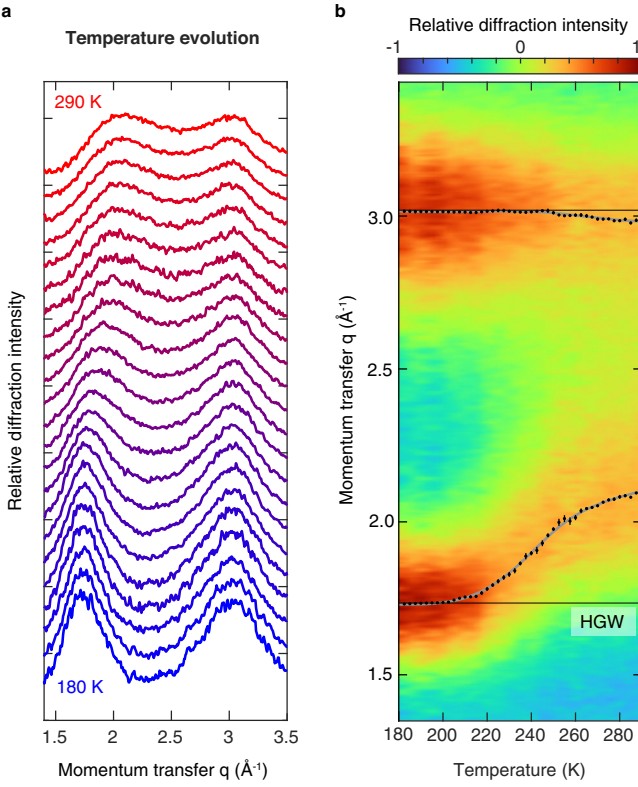

**Fig. 3 | Structural evolution of water in no man's land. a** Diffraction patterns reveal that the structure of water evolves smoothly as it is cooled from 290 K to 180 K. Diffraction patterns are shown in 5 K steps. **b** Two-dimensional plot of the evolution of the diffraction pattern of water. Black dots mark the positions of the first two diffraction maxima, with the error bars indicating to the standard error of the mean of five measurements. The gray lines provide a guide to the eye and are derived from splines (Supplementary Methods G). The horizontal black lines indicate the positions of the diffraction maxima of Hyperquenched Glassy Water (HGW). Source data are provided as a Source Data file.

diffraction maximum exhibits a somewhat smaller temperature dependence in our experiment, which is likely due to an overlap with a diffraction feature arising from the graphene support (Supplementary Fig. 6). We therefore do not include the evolution of the second diffraction maximum in the interpretation of our data.

The position of the first diffraction maximum of water exhibits an s-shaped temperature evolution. Upon cooling the room temperature liquid, the maximum shifts to lower momentum transfer, with the shift accelerating below about 250 K. Just below 200 K, the peak position converges to that of hyperquenched glassy water (HGW, horizontal lines in Fig. 3b), a form of amorphous ice that is formed when liquid water is cooled at rates exceeding $10^6$ K/s (Ref. [20]) and that we obtain in our experiment when we simply switch off the laser to let the liquid cool at maximum speed.

Our data reveal a continuous evolution of the structure of water as the liquid is cooled, which only slows once it approaches the structure of HGW below 200 K. We therefore infer that at higher temperatures, the liquid fully relaxes on the timescale of our experiment. If this were not the case, we would observe that the structure of the liquid would get arrested at a higher temperature. In contrast, it continues to evolve until it has converged to that of HGW below 200 K. This is consistent with previous studies, which show that the relaxation time of water exceeds the timescale of our experiment (~ 5 μs) only at temperatures below 185 K (Ref. [21]). Our data thus confirm that water in no man's land can be equilibrated before crystallization occurs, a point that had previously been debated[6,11,22].

The smooth evolution of the diffraction pattern of water appears inconsistent with theories that predict the liquid to undergo a first-order phase transition under our experimental conditions. In this case, one would expect to observe a signature in the temperature dependence of the structure factor[23]. Instead, the evolution of water from its room-temperature to its low-temperature structure occurs continuously over a wide temperature interval of about 40 K, between 220 K and 260 K.

In the liquid-liquid critical point scenario, the anomalies of water arise as the liquid approaches the Widom line, which represents the locus of the maximum rate of change of its properties. Studies on transiently heated ice films find a maximum rate of change at 210 K (Ref. [12]). However, the authors note that the experiment is not sensitive to structural changes above 230 K, so that this temperature cannot be identified with the crossing of the Widom line. X-ray diffraction experiments on microdroplets have deduced a temperature of maximum change of 229 K (Ref. [19]). Our analysis has the benefit of including the entire temperature range. Interestingly, it places this temperature at $243 \pm 2$ K, where the evolution of the first diffraction maximum has an inflection point. Taking into account the temperature evolution of the high- and of the low-temperature liquid, one can estimate the location of the hypothetical Widom line to occur at $234 \pm 1$ K using the procedure suggested in Ref. [24] (see Supplementary Note J).

By capturing the structural evolution of water throughout no man's land, our experiments help narrow down the range of possible explanations for the origin of the water anomalies and provide a stringent test for the development of accurate water models. Experiments at higher pressure will be required in order to test the remaining scenarios, which include explanations that do not invoke a singularity[3]. Our approach for rapidly preparing water in no man's land and equilibrating it at a well-defined temperature should be quite general. For example, it will be straightforward to combine it with a range of different probes, such as infrared, Raman[25], or x-ray absorption spectroscopy, which are each sensitive to different properties of water. Moreover, our approach opens up new avenues for studying the dynamics of supercooled water, with preliminary experiments showing that we can obtain insights into the crystallization process. Finally, our experiments also bear relevance to cryo-electron microscopy, which appears set to become the preferred method in structural biology[26]. In cryo-electron microscopy, vitrified protein samples are prepared through hyperquenching. Our experiments capture the structural evolution of water during vitrification and therefore promise to shed new light onto the question of how well the process is able to preserve the room-temperature structure of proteins[27].

## Data availability

Positions of the diffraction maxima of water are included as a Source Data file. All other data supporting the findings of the study are available from the corresponding author upon request. Source data are provided with this paper.

## Code availability

Computer code used to generate the results of this paper are available from the corresponding author upon request.

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

## Acknowledgements
The authors would like to thank Dr. Pavel K. Olshin for his help with the heat transfer simulations as well as Dr. Jonathan M. Voss for his help with the preparation of Fig. 1. The authors would also like to thank Dr. Greg Kimmel for pointing out the method for estimating the location of the hypothetical Widom line in Ref. 24. This work was supported by the ERC Starting Grant 759145 and by the Swiss National Science Foundation Grant PP00P2_163681, both awarded to U.J.L.

## Author contributions
U.J.L. was responsible for conceptualizing this work. The methodology, investigation and visualization was done by C.R.K., N.J.M., G.B., and U.J.L. Acquiring funding, project administration and supervision was performed by U.J.L. The writing of the original draft was performed by C.R.K., N.J.M., G.B., and U.J.L. The review and editing of the manuscript was performed by all authors.

## Competing interests
The authors declare no competing interests.
