## [Peer Review File · Nature Communications]

Editorial Note: Parts of this peer review file have been redacted as indicated to maintain the confidentiality of other journals.

REVIEWER COMMENTS

Reviewer #1 (Remarks to the Author):

I reviewed this manuscript when it was initially submitted to [redacted]. At the time, I wrote (in part): "The novelty of the current experiments is that they provide information on equilibrated water over a much wider temperature range than previously available. The key result is that structure of equilibrated water continuously evolves from one characteristic of a high-density liquid at high temperatures to a low-density liquid at low temperatures. The experiments are carefully done, the results are clearly presented, and the data justify the conclusions. Because water is a key component in so many fields, understanding its fundamental properties is of interest to a broad range of scientists. Therefore, I believe this research is suitable for publication in [redacted]."

In the revised manuscript, the authors have thoroughly addressed the issues identified in the initial review. Therefore, I recommend publication of the revised article in *Nature Communications*".

Reviewer #2 (Remarks to the Author):

I have reviewed the revised manuscript and supplementary information. I am happy with the responses of the authors to both my previous report and that of the other referee. I have no further comments to make and believe this paper fits well within the scope of *Nature Communications*.

Reviewer #3 (Remarks to the Author):

I have no doubt the authors have provided a new experimental tool to investigate supercooled water at lower temperatures than previously achieved. However, the paper lacks an in-depth scientific discussion on the impact of the observation on the current debate of the two-liquid structure model of water. I still find the citations to previous works were selective and not comprehensive.

The only new result is the observation of a S-shaped curve in the FSDP when water is cooled. It was speculated that the inflection point might be due to the crossing of the Widom line. However, it has been pointed out by the first reviewer that the feature is not definitive. Measurements under pressure are required to confirm it. If the position of the FSDP can be taken as a thermodynamic order-parameter, the inflection point may be interpreted as a second-order transition from a water-like (HDA?) structure to LDA (ice-like) as well. Note that the transition of HDA prepared from compressed ice to LDA is a multi-step process (Tulk, *Science*, 297, 1320 (2002)). Substantial heat is released associated with transforming from a high-density solid to a low-density form. The evolved heat will be absorbed quickly, leading to a fast, first-order transition. Furthermore, a high-density form of ice can be prepared by rapid quenching at temperatures < 30 K (Narten, *J. Chem. Phys.*, 64, 1106 (1976); Jennissen, *Sci.*, 265, 753; Jennissen, *Astrophys. J.* 455, 389 (1995) with diffraction features resemble that of liquid water. Upon heating, the high-density form converted to LDA, which has a structure close to ice Ic. Note FSDP of 1.73 Å⁻¹ on the sample measured below 200 K reported in this paper is almost identical to the previously reported. Moreover, the structure of ASW and LDA are almost identical (H. Li, *JPCB*, 125, 13320 (2021)). Therefore, it is not unreasonable to attribute the onset of the high-density to low-density transformation starting at ca 260 K and completed at 200K. The narrowing of the line widths below 200 K mentioned in my last report supports this interpretation.

The authors contributed to a significant technical improvement in the study of ultrafast electron diffraction and extended the measurements to 100K. Unfortunately, the discussion was not detail enough or unequivocal in helping to resolve the current problems in amorphous ice / water. The paper in its present form is not accepted due to the stringent criteria of *Nat. Comm.*

Referee #1

I reviewed this manuscript when it was initially submitted to [redacted]. At the time, I wrote (in part): "The novelty of the current experiments is that they provide information on equilibrated water over a much wider temperature range than previously available. The key result is that structure of equilibrated water continuously evolves from one characteristic of a high-density liquid at high temperatures to a low-density liquid at low temperatures. The experiments are carefully done, the results are clearly presented, and the data justify the conclusions. Because water is a key component in so many fields, understanding its fundamental properties is of interest to a broad range of scientists. Therefore, I believe this research is suitable for publication in [redacted]."

In the revised manuscript, the authors have thoroughly addressed the issues identified in the initial review. Therefore, I recommend publication of the revised article in Nature Communications".

Referee #2

I have reviewed the revised manuscript and supplementary information. I am happy with the responses of the authors to both my previous report and that of the other referee. I have no further comments to make and believe this paper fits well within the scope of Nature Communications.

Referee #3

I have no doubt the authors have provided a new experimental tool to investigate supercooled water at lower temperatures than previously achieved. However, the paper lacks an in-depth scientific discussion on the impact of the observation on the current debate of the two-liquid structure model of water. I still find the citations to previous works were selective and not comprehensive.

The only new result is the observation of a S-shaped curve in the FSDP when water is cooled. It was speculated that the inflection point might be due to the crossing of the Widom line. However, it has been pointed out by the first reviewer that the feature is not definitive. Measurements under pressure are required to confirm it. If the position of the FSDP can be taken as a thermodynamic order-parameter, the inflection point may be interpreted as a second-order transition from a water-like (HDA?) structure to LDA (ice-like) as well. Note that the transition of HDA prepared from compressed ice to LDA is a multi-step process (Tulk, Science, 297, 1320 (2002)). Substantial heat is released associated with transforming from a high-density solid to a low-density form. The evolved heat will be absorbed quickly, leading to a fast, first-order transition. Furthermore, a high-density form of ice can be prepared by rapid quenching at temperatures < 30 K (Narten, J. Chem. Phys., 64, 1106 (1976); Jenniskanen, Sci., 265, 753; Jenniskanen, Astrophys. J. 455, 389 (1995) with diffraction features resemble that of liquid water. Upon heating, the high-density form converted to LDA, which has a structure close to ice Ic. Note FSDP of 1.73 Å⁻¹ on the sample measured below 200 K reported in this paper is almost identical to the previously reported. Moreover, the structure of ASW and LDA are almost identical (H. Li, JPCB, 125, 13320 (2021)). Therefore, it is not unreasonable to attribute the onset of the high-density to low-density transformation starting at ca 260 K and completed at 200K. The narrowing of the line widths below 200 K mentioned in my last report supports this interpretation.

The authors contributed to a significant technical improvement in the study of ultrafast electron diffraction and extended the measurements to 100K. Unfortunately, the discussion was not detailed enough or unequivocal in helping to resolve the current problems in amorphous ice / water. The paper in its present form is not accepted due to the stringent criteria of Nat. Comm.

Indeed, experiments at higher pressure will be required to definitively distinguish between the remaining scenarios that our data are consistent with. As the reviewer points out correctly, this includes scenarios that do not invoke a singularity. We have explicitly added these points to our manuscript. We feel however that it is beyond the scope of our manuscript to go into further details about the many scenarios that have been put forward over the years. Instead, we have added a reference to what is widely considered the authoritative review on the topic.